# Adaptive Mesh-Quantization for Neural PDE Solvers

**Winfried van den dool**                                          *w.v.s.o.vandendool@uva.nl*
*QUVA Lab*
*University of Amsterdam*
**Maksim Zhdanov**                                                *m.zhdanov@uva.nl*
*AMLAB*
*University of Amsterdam*
**Yuki M. Asano**                                                 *yuki.asano@utn.de*
*FunAI Lab*
*University of Technology Nuremberg*
**Max Welling**                                                   *m.welling@uva.nl*
*AMLAB*
*University of Amsterdam*

Reviewed on OpenReview: *https://openreview.net/forum?id=NN17y897WG*

## Abstract

Physical systems commonly exhibit spatially varying complexity, presenting a significant challenge for neural PDE solvers. While Graph Neural Networks can handle the irregular meshes required for complex geometries and boundary conditions, they still apply uniform computational effort across all nodes regardless of the underlying physics complexity. This leads to inefficient resource allocation where computationally simple regions receive the same treatment as complex phenomena. We address this challenge by introducing Adaptive Mesh Quantization: spatially adaptive quantization across mesh node, edge and cluster features, dynamically adjusting the bit-width used by a quantized model. We propose an adaptive bit-width allocation strategy driven by a lightweight auxiliary model that identifies high-loss regions in the input mesh. This enables dynamic resource distribution in the main model, where regions of higher difficulty are allocated increased bit-width, optimizing computational resource utilization. We demonstrate our framework's effectiveness by integrating it with two state-of-the-art models, MP-PDE and GraphViT, to evaluate performance across multiple tasks: 2D Darcy flow, large-scale unsteady fluid dynamics in 2D, steady-state Navier–Stokes simulations in 3D, and a 2D hyper-elasticity problem. Our framework demonstrates consistent Pareto improvements over uniformly quantized baselines, yielding up to 50% improvements in performance at the same cost.

## 1 Introduction

In recent years, there has been a surge in the application of machine learning algorithms to build neural surrogates for Partial Differential Equations (PDEs). Using neural networks, PDEs can be modeled as a next-frame prediction problem, where the dynamics of the system (such as fluid movements) are implicitly learned. In real-world applications, traditional numerical methods such as the finite element method predominantly operate on meshes or irregular grids (Mavriplis, 1997), as these representations naturally adapt to complex geometries. Graph Neural Networks (GNNs) (Brandstetter et al., 2022; Pfaff et al., 2020; Gilmer et al., 2017b) have emerged as a natural solution, since they can operate directly on unstructured meshes, thereby avoiding the mesh-to-grid interpolation otherwise needed to represent irregular domains on regular grids. However, since computation scales directly with input graph size, efficient inference of GNNs remains challenging (Zhang et al., 2022). Furthermore, reducing the operational cost is crucial for the practical deployment of GNN-based neural surrogates, as lower costs enable higher data resolution - a key factor in improving modeling accuracy (Iles et al., 2020; Bauer et al., 2015; Randall et al., 2007).

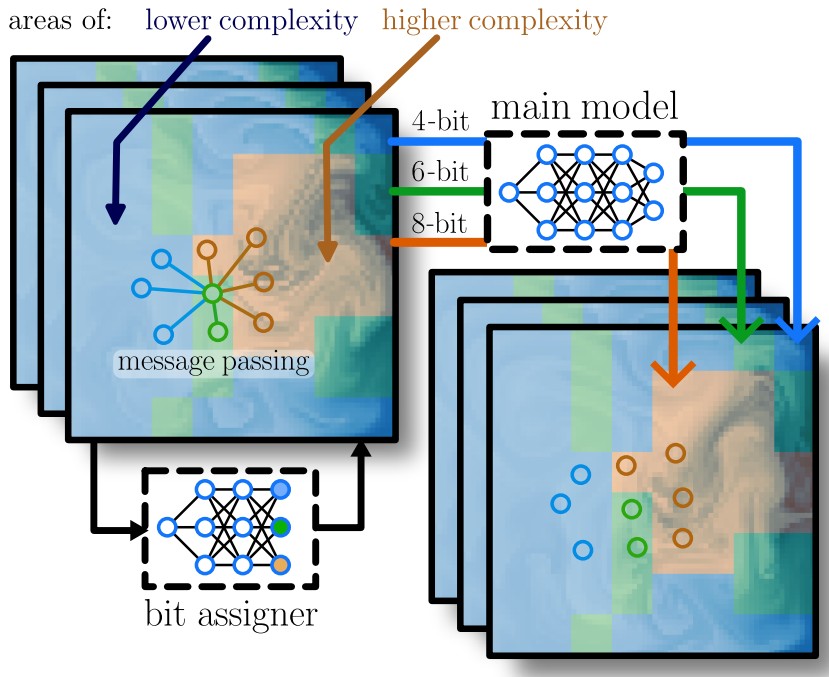

Figure 1: Overview of the proposed framework. Given a point cloud or graph, a bit assigner returns a node-wise quantization scheme for a larger model. The goal is to assign higher precision to more difficult and complex regions.

A common way to lower the computational budget required for inference in deep learning is (integer) quantization (Jacob et al., 2017). Converting model weights and activations from floating-point to integer representations can drastically reduce computing and memory requirements. Although quantization was initially associated with resource-limited devices, it is now gaining attention in broader applications including Large Language Models (Wang et al., 2023) and large-scale physical simulations (Lang et al., 2021). In the latter, reduced-precision arithmetic has shown particular promise in mathematical (climate) modeling (Kimpson et al., 2023) and neural surrogates (Dool et al., 2023), enabling higher data resolutions within fixed computational budgets.

While quantization typically applies uniform precision reduction across the entire model, many physical systems exhibit complexity that varies in both location and time. Phenomena such as turbulence, shock waves, and steep gradients create regions where accurate prediction requires substantially higher numerical precision than in other, smoother, more predictable areas.(Strogatz, 2000) Current neural PDE solvers treat all mesh nodes uniformly, allocating the same computational resources regardless of the underlying modeling difficulty at each location. This uniform approach leads to inefficient resource utilization, where computationally straightforward regions consume the same resources as challenging phenomena.

We address this inefficiency by introducing Adaptive Mesh Quantization (AMQ). In mesh quantization, we apply quantization to all the mesh-based features - including node features, edge features, and possibly cluster representations - processed by graph neural networks operating on meshes. Our adaptive approach dynamically varies this quantization - allocating higher bit-width to regions of higher complexity.

More concretely, relatively easy nodes - as to be selected by a small auxiliary model - will be processed using low bit-width, freeing up resources for more complex regions of the input requiring high precision (Figure 1). The light-weight auxiliary model is trained to predict the loss of the main model at each node, which serves as a proxy for local complexity. This proxy is thus based on both aleatoric and epistemic uncertainty, signaling not only where prediction is hard in general but also where prediction is hard for the quantized main model specifically.

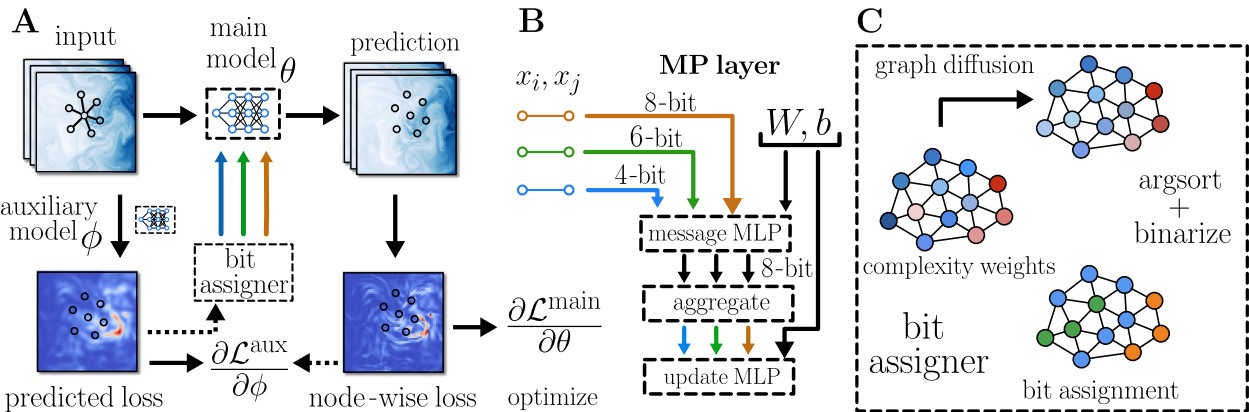

Figure 2: Overview of the training phase (**A**). A lightweight auxiliary model assigns a complexity weight to every mesh node to guide the resource allocation of the large main message-passing model. Gradients do not flow through the dotted black arrows, i.e., backpropagation happens through the solid black arrows. (**B**). The bit assignment process is schematically shown in (**C**).

Overall, we make the following contributions:

- We introduce Adaptive Mesh Quantization, a framework that learns to allocate resources locally on a mesh depending on the input data. We additionally propose an efficient implementation for the mixed-precision quantization of MLPs.

- We investigate the particular setup where an auxiliary model learns which input regions are more complex based on the main model's loss landscape, training both models simultaneously.

- We validate the framework on multiple PDE datasets, ranging in complexity and scale, and demonstrate consistent compute-performance Pareto improvements.

## 2 Related work

### 2.1 Neural PDE solvers

The intersection of PDE solving and deep learning has emerged as an active research area (McGreivy & Hakim, 2024). The usual task is to learn a mapping from an initial state to a solution of a PDE with a neural network, in order to accelerate complex simulations in fields like fluid dynamics and climate modeling (Wang et al., 2024). Originally focusing on regular grids (Li et al., 2020), multiple solutions were suggested to deal with irregular meshes, with message-passing-based approaches (Brandstetter et al., 2022) most relevant to our work. For these networks the input comes in the form of a graph and a model learns evolution for every node. Lately, mesh attention (Janny et al., 2023) was also introduced to capture long-range dependencies further enhancing the performance of mesh-based models.

We do not introduce a new pde-solving model, but rather a new method that is applicable to families of Graph Neural Networks in particular. To showcase our method we use both the original MPNN (Brandstetter et al., 2022), as well as the more elaborate Graph Attention architecture (Janny et al., 2023). Apart from being state-of-the-art on several datasets, the latter model has the advantage of covering many different layer types (e.g., graph pooling, rnn, attention), ensuring our method is broadly applicable across different GNN model architectures.

## 2.2 Quantization

By reducing the precision of storage and arithmetic operations in neural networks, quantization offers significant reductions in memory overhead and computational costs while typically maintaining model performance (Jacob et al., 2017; Nagel et al., 2021). Conventional approaches implement uniform quantization across all model layers, either through post-training quantization (PTQ) (Sung et al., 2015) or during the training phase via Quantization-Aware Training (QAT) or Quantized Training (QT) (Jacob et al., 2017; Nagel et al., 2021). Mixed-precision quantization (Wang et al., 2018; Pandey et al., 2023) has emerged as a more efficient approach, allowing different layers or operations to use different numerical precisions. Automated frameworks for bit-width selection (Wu et al., 2018) have shown that optimized quantization strategies can achieve higher compression rates while preserving model accuracy.

Of particular relevance to our work are the quantization of PDE solvers (van den Dool et al., 2023) and input-dependent quantization. van den Dool et al. (2023) show that quantization can be effectively applied to neural PDE solvers, and importantly, that spatial resolution modification and quantization represent complementary approaches that can be combined for optimal results.
In input-dependent quantization, Liu et al. Liu et al. (2022) and Saxena & Roy (2023) employ auxiliary models to dynamically assign bit-widths to neural network layers based on input complexity, thereby achieving superior model performance while maintaining the benefits of quantization. In contrast to these works, our method applies dynamic mixed quantization not per input instance, but on the complexity of particular areas *within* a single input.

## 2.3 Message-passing quantization

Quantization methods have recently gained popularity for MPNNs in particular, as inefficient inference is a known problem when scaling MPNNs up to real-world and large-scale graph applications (Ma et al., 2024). Tailor et al. (2020) propose a QAT method which selectively quantized nodes with low in-degree as less prone to the quantization noise. Zhu et al. (2023) further advanced the idea by learning the quantization bandwidth for each node based on their in-degree. However, these methods are based on the graph architecture alone, which may not always be related to, or take into account, the complexity of the data that it represents.

# 3 Preliminaries

## 3.1 Quantization

To move from floating-point to efficient fixed-point operations, two quantization parameters are needed: the scale factor $s$, and the bit-width $b$. Together these parameters form a *quantizer* $Q(\cdot) = Q(\cdot\,; s, b)$ which acts on a floating point tensor $\mathbf{x}$ as

$$Q(\mathbf{x}) = \text{clamp}\left(\lfloor s \cdot \mathbf{x} \rceil; -2^{b-1} + 1, 2^{b-1} - 1\right), \tag{1}$$

where $\lfloor \cdot \rceil$ is the round-to-nearest operator and $\text{clamp}(\cdot, l, h)$ clamps an input between the values $l$ and $h$. The approximate real value of $x$ is then retrieved by

$$\hat{\mathbf{x}} \;=\; \frac{Q(\mathbf{x})}{s} \approx \mathbf{x}. \tag{2}$$

Note that we do not apply shift, which helps us preserve the 0 value. Here $\mathbf{x}$ can represent both a weights matrix and an activation vector. The bit-width is chosen before training, thereby fixing the target range to $[-2^{b-1} + 1, 2^{b-1} - 1]$. The scale factor $s$ ensures efficient use of this target range, using statistics of $\mathbf{x}$ like $\max|\mathbf{x}|$ in the most basic case. If $\mathbf{x}$ are activations, exponential moving averages of such statistics can be tracked during training, to be fixed after an initial calibration training phase has been completed. The scale must be chosen to balance outlier clipping and rounding errors. To reduce such errors, $s$ is often defined on a per-channel basis, which can be seen as replacing $s$ with diagonal matrices $\mathbf{S}$ and $\mathbf{S}^{-1}$ in Equations 1 and 2, respectively.

The efficiency gains from quantization in a multiplication like $\hat{\mathbf{W}} \cdot \hat{\mathbf{a}}$ for weights matrix $\mathbf{W}$ and activations $\mathbf{a}$ follows when taking the $\frac{1}{s}$ out of the matrix multiplication:

$$\hat{\mathbf{W}} \cdot \hat{\mathbf{a}} = \frac{1}{s_W s_a} \left( Q(\mathbf{W}) Q(\mathbf{a}) \right) \approx \mathbf{W} \cdot \mathbf{a}. \tag{3}$$

The cost of computing $(Q(\mathbf{W}) Q(\mathbf{a}))$ quantized using $b_W$, $b_a$ bits is then in the order of $b_W \cdot b_a$ Multiply Accumulate operations (MACs).[1] As the bias vector's impact on computational cost is commonly considered negligible, it is kept in floating point precision.

### 3.2 Message-passing neural networks

The neural network input is a graph $G = (\mathcal{V}, \mathcal{E})$, which comes from a discretization of the $D$-dimensional computational domain $\Omega \in \mathbb{R}^D$. It might, for example, represent a mesh or a point cloud with induced connectivity (e.g. by $k$-nearest neighbours) and contains $N$ nodes. Here $\mathcal{V} = \{1, ..., N\}$ is the set of nodes and $\mathcal{E} \subseteq \mathcal{V} \times \mathcal{V}$ are the edges. Additionally, every node is endowed with its location $\mathbf{p_i} \in \Omega$ and, optionally, features $\mathbf{x}_i \in \mathbb{R}^d$. Message-passing neural networks are architectures developed specifically to operate on graph-structured data (Satorras et al., 2021; Brandstetter et al., 2022). Following the notation from Gilmer et al. (2017a), we define the $l$-th layer as a sequence of operations:

$$
\begin{aligned}
\mathbf{m}_{ij} &= \mathrm{MLP}_e(\mathbf{x}_i^l, \mathbf{x}_j^l, \mathbf{p}_i - \mathbf{p}_j), & \text{message} \\
\mathbf{m}_i &= \sum_{j \in \mathcal{N}(i)} \mathbf{m}_{ij}, & \text{aggregate} \\
\mathbf{x}_i^{l+1} &= \mathrm{MLP}_n(\mathbf{x}_i^l, \mathbf{m}_i), & \text{update}
\end{aligned}
\tag{4}
$$

where $\mathcal{N}(i)$ represents the set of neighbours of node $v_i$, $\mathrm{MLP}_e, \mathrm{MLP}_n$ are message (edge) and update (node) MLPs respectively, and $\mathbf{x}_i^l \in \mathbb{R}^d$ are latent $d$-dimensional feature vectors.

MP-PDE (Brandstetter et al., 2022) follows the Encode-Process-Decode framework of Sanchez-Gonzalez et al. (2020) with the core element being $M$ steps of learned message passing in the processor. More precisely, the processor is an MPNN with additional input (solution difference) in the message network:

$$\mathbf{m}_{ij} = \mathrm{MLP}_e(\mathbf{x}_i^l, \mathbf{u}_i - \mathbf{u}_j, \mathbf{x}_j^l, \mathbf{p}_i - \mathbf{p}_j), \qquad \text{message} \tag{5}$$

where $\mathbf{u}$ is a vector of solution values at previous time steps. The encoder and decoder are applied node-wise and implemented using an MLP and a shallow 1D convolutional network, respectively.

Janny et al. (2023) leverages recent developments in computer vision to produce a scalable mesh transformer. The main idea is to capture long-term interactions in the domain, for which the mesh is coarsened and multi-head attention (Vaswani et al., 2017) is computed on the clusters. The final model (GraphViT) consists of 5 steps: 1) MPNN encoder (Eq. 4), 2) clustering, 3) graph pooling to the clusters, 4) attention, and 5) MPNN decoder. In our experiments, we found that steps 1 and 5 are by far the most computationally demanding in the model (see Appendix C for the details), which makes the model suitable for our framework to test its scalability.

---

[1]It is common to use MACs to represent compute cost, rather than wall-time. This is partly because wall-time is hardware-dependent, but also because often quantization *simulation* packages are used, as optimal on-device implementation is not possible or reserved for real-world applications (Siddegowda et al., 2022).

## 4 Method

### 4.1 Adaptive Mesh Quantization with fixed budget

We assume that for an ordered set $\mathcal{V}$ of $N$ nodes a weight $\mathbf{w} \in \mathbb{R}^N$ is given that relates to the complexity of (predicting the output at) each node. We further assume a fixed total computational budget which we want to distribute across different quantization levels efficiently at inference time based on those complexity weights $\mathbf{w}$.

Let $Q$ be the used quantizers for different bit-widths, as defined in Equation 1, so that $Q_k$ has quantization parameters given by bit-width $b_k$ and scale $s_k$, and let $k$ be increasing in compute cost. For example, for a setup with `Int4`, `Int8` and `Int12` quantization, $Q = (Q_1, Q_2, Q_3)$ with $b = (4, 8, 12)$. Let associated budget allocation ratios $\boldsymbol{\alpha} \geq 0$ be given such that $\sum_k \boldsymbol{\alpha}_k = 1$.

To map the complexity weights to quantization levels we first sort the weights vector $\mathbf{w}$, and then proceed by assigning the most expensive (i.e., having the highest bit-width) quantizer $Q_k$ to the $\boldsymbol{\alpha}_k$ fraction of nodes with the highest weight, the second-highest $Q_{k-1}$ to the next $\boldsymbol{\alpha}_{k-1}$ fraction, and so forth, ensuring that nodes with higher weights receive more precision, as shown in Algorithm 1.

---

**Algorithm 1 Weights-based Quantization Assignment**

1: **Input:** Complexity weights $\mathbf{w} \in \mathbb{R}^N$ Available quantization functions $Q = [Q_1, ..., Q_K]$ Allocation ratios $\boldsymbol{\alpha} = [\boldsymbol{\alpha}_1, ..., \boldsymbol{\alpha}_K]$ with $\sum_i \boldsymbol{\alpha}_i = 1, \boldsymbol{\alpha}_i \geq 0$
2: $I \leftarrow \text{argsort}(\mathbf{w})$ {Sort by increasing weight}
3: $\boldsymbol{\beta} \leftarrow [\emptyset] \times K$ {Initialize empty buckets}
4: start $\leftarrow 0$
5: **for** $i = 1$ to $K$ **do**
6:     end $\leftarrow$ start $+ \lfloor N \cdot \boldsymbol{\alpha}_i \rfloor$
7:     $\boldsymbol{\beta}_i \leftarrow I[\text{start} : \text{end}]$ {Assign indices to bucket}
8:     start $\leftarrow$ end
9: **end for**
10: **Output:** $\boldsymbol{\beta}$

---

The actual bit-width of node $i$ is effectively used to quantize its activations in the update MLP of Equation 4 and the embedding and projection layers at the start and end of the GNN architecture. In Section 4.4 we describe how MLPs can be efficiently implemented to handle a general multi-resolution bit assignment for all nodes.

We also use the node weights $\mathbf{w}$ to assign a similar complexity proxy weight to the edges and clusters (if available) by using the target node weight for the edges and the mean weight of all cluster nodes, respectively. For both edges and clusters, we also follow the Alg. 1 to get a bit assignment that follows the same budget allocation ratios $\boldsymbol{\alpha}$. The bit-width of edge $j \rightarrow i$ is used to quantize the activations of the message MLP of Equation 4 and possibly edge embedding and update layers if included, whereas the cluster bits assignment is similarly used in the Gated Recurrent Unit of the RNN pooling layer and processing MLPs. For further architecture details, we refer to the supplementary material.

For each MLP, only the activations are adaptively and non-uniformly quantized. The weights are quantized to fixed bit-width, e.g., `Int8` in our case, independent of the presumed complexity of the input elements. Assuming a fixed number of nodes, edges and clusters, our use of a fixed budget allocation ratio $\boldsymbol{\alpha}$ not only directly translates to a fixed computational budget, but also enables the design of a predefined model architecture on hardware.

### 4.2 Predicting spatial complexity

Let $G = (\mathcal{V}, \mathcal{E})$ be the model input graph with node features $\mathbf{x} \in \mathbb{R}^{N \times d}$, where $N$ is the number of nodes and $d$ is the feature dimension, and let $\mathbf{y} \in \mathbb{R}^{N \times d'}$ be the target outputs. We aim to assign a weight $\mathbf{w}_i > 0$ to each node $i$, representing its complexity, using a small auxiliary GNN model denoted by $A_\phi : (G, \mathbf{x}) \rightarrow \mathbf{w}$.

To enforce the interpretation of $\mathbf{w}$ as representing the node complexity, we propose to train the auxiliary model $A_\phi$ to predict the loss of a larger main model $M_\theta : (G, \mathbf{x}) \to \hat{\mathbf{y}}$. Note that we ultimately seek a good model $M_\theta$, but we first explain how $A_\phi$ is trained. We define $L_M \in \mathbb{R}^N_{\geq 0}$ as the loss of the main model which has not been spatially reduced, i.e., if $\mathcal{L}_M(M_\theta(\mathbf{x}), \mathbf{y})$ is the actual loss of this model for input $\mathbf{x}$, then

$$\mathcal{L}_M(M_\theta(\mathbf{x}), \mathbf{y}) = \frac{1}{N} \sum_{i=1}^{N} L_M(M_\theta(\mathbf{x}), \mathbf{y})[i], \tag{6}$$

where $[i]$ means selecting the $i$-th node, omitting the dependence on $G$ for clarity. If a conventional loss function is used (e.g, MSE) $L_M[i]$ measures the error between the predicted value and the ground truth value of node $i$. We use this error for our proxy of node complexity: After applying a smoothing function $S : \mathbb{R}^N \to [0, 1]^N$ to the spatial loss $L$, we use the smoothed result as a target for the auxiliary model $A_\phi$.

More precisely, the parameters $\phi$ are updated using Stochastic Gradient Descent (SGD) with learning rate $\eta_A$ as:

$$
\begin{aligned}
L_M &= L_M(M_\theta(\mathbf{x}), \mathbf{y}), \\
\phi_{t+1} &= \phi_t - \eta_A \nabla_\phi \mathcal{L}_A \Big( A(\mathbf{x}; \phi_t), S(L_M) \Big).
\end{aligned}
\tag{7}
$$

The smoothing function ensures that the target for the auxiliary model training is sufficiently well-behaved. It involves the following steps:

1. **Normalization**. We scale the loss $L_M$ by setting $L_M \leftarrow L_M / \max_i L_M[i]$. This transformation normalizes $L_M$ to the $[0, 1]$ range, which both stabilizes the loss distribution and mitigates the influence of extreme values on auxiliary model gradients. It also allows the auxiliary model $A$ to use a sigmoid activation in its final layer, simplifying training.

2. **Graph Diffusion**. We apply multiple rounds of graph diffusion (see Appendix, Equation 15) so that loss information propagates across neighboring nodes. The underlying assumption is that the loss observed at any given node reflects not only its own operations but also the influence of nearby nodes, as a result of the message-passing framework. By performing diffusion steps, we effectively smoothen the loss values over the graph, reflecting this interdependence. The diffusion process smooths the loss landscape and further reduces the effect of local outliers.

### 4.3 Training

Instead of updating the auxiliary model on a fixed (e.g., pre-trained) model $M$, as suggested in equation 7, we propose a single synchronous training procedure, using a framework that couples the auxiliary model $A_\phi$ and the main model $M_\theta$. This coupling is based on:

- Applying Alg. 1 to the proxy loss outputs $\mathbf{w}$ of $A_\phi$ to obtain a bit-width assignment $\boldsymbol{\beta}$; where $\boldsymbol{\beta}$ denotes a combined representation for the bit-width assignment of the nodes, edges and clusters:

$$
\begin{aligned}
\mathbf{w} &= A(\mathbf{x}; \phi_t), \tag{8} \\
\boldsymbol{\beta} &= \text{AssignQuant}(\mathbf{w}, Q, \boldsymbol{\alpha}), \tag{9}
\end{aligned}
$$

with $Q$ and $\boldsymbol{\alpha}$ the available quantization functions and allocation ratios, as defined in Section 4.1.

- Quantizing the main model as described in Section 4.1. We now define the main model $M$ as

$$M_{\theta, \beta} : (G, \mathbf{x}) \to \hat{\mathbf{y}}. \tag{10}$$

- Using the spatial loss of this quantized model as $L_M$ in equation 7:

$$L_M = L_M(M_{\theta, \beta}(\mathbf{x}), \mathbf{y}). \tag{11}$$

---

**Algorithm 2 Basic mixed-precision linear layer**

---

**Input:** activations $\mathbf{a} \in \mathbb{R}^{N \times d}$, weight matrix $\mathbf{W}^T \in \mathbb{R}^{d \times d'}$, available quantization functions $Q = [Q_1, ..., Q_K]$, index buckets $B = [B_1, \ldots, B_K]$
$\{\mathbf{a}_k\}_{k=1}^K \leftarrow \text{Distribute}(\mathbf{a}, B)$     $\{\text{Get } \mathbf{a}_k \in \mathbb{R}^{|B_k| \times d}\}$
Initialize $Y \leftarrow [\emptyset]$
**for** $k = 1$ to $K$ **do**
   $\mathbf{y}_k \leftarrow Q_k(\mathbf{a}_k) \cdot Q^w(\mathbf{W}^T)$
   $\mathbf{y}_k \leftarrow \frac{\mathbf{y}_k}{s_k s_W}$
   $Y \leftarrow Y \cup \{\mathbf{y}_k\}$     $\{\text{append } \mathbf{y}_k \text{ to } Y\}$
**end for**
$Y \leftarrow \text{Concatenate}(Y)$     $\{\text{unpack buckets into one array}\}$
$Y \leftarrow \text{Reorder}(Y)$     $\{\text{place in original order}\}$
**Output:** $Y$

---

Before the actual training, we may start with an initial $\mathbf{w}_0$, potentially based on data uncertainty, reflecting the prior estimate of complexity on the mesh.[2] We then use SGD with learning rate $\eta$ to update the parameters $\phi$ and $\theta$ as follows:

$$
\begin{aligned}
\mathbf{w}_{t+1} &= A(\mathbf{x}; \phi_t) \\
\boldsymbol{\beta}_{t+1} &= \text{AssignQuant}(\mathbf{w}_{t+1}, Q, \boldsymbol{\alpha}), \\
\theta_{t+1} &= \theta_t - \eta_M \nabla_\theta \mathcal{L}_M \big( M(\mathbf{x}; \boldsymbol{\beta}_{t+1}, \theta_t), \mathbf{y} \big), \\
\phi_{t+1} &= \phi_t - \eta_A \nabla_\phi \mathcal{L}_A \Big( A(\mathbf{x}; \phi_t), L_M \big( M(\mathbf{x}; \boldsymbol{\beta}_{t+1}, \theta_t), \mathbf{y} \big) \Big).
\end{aligned}
\tag{12}
$$

By basing our node complexity proxy $A(\mathbf{x}; \phi_t)$ on the loss of an existing model, instead of on the underlying data alone, we hypothesize that it better reflects epistemic (model) uncertainty, instead of just aleatoric (dataset) uncertainty. Training with an already quantized model, $M(\mathbf{x}; \boldsymbol{\beta}_t, \theta_t)$, in the loop then allows the modeling complications caused by quantization to be reflected in the complexity proxy that we use, providing direct feedback to the main model's precision requirements.

### 4.4 Hardware-efficient multiplications with non-uniform quantization

In this section, we describe how a non-uniform assignment of activation bit-widths can be used in a single quantized layer. Let $B = [B_1, ..., B_K]$ be the buckets of indices of vectors quantized using $Q = [Q_1, ..., Q_K]$. The quantizers are defined as in Equation 1 and implicitly include their corresponding quantization parameters, i.e., the bit-width $b_k$ and scale factor $s_k$.

Let the weight matrix of a linear layer be given by $\mathbf{W}^T \in \mathbb{R}^{d \times d'}$, and assume it gets quantized using scale parameter $s_W$ and bit-width $b_W$. Let $\mathbf{a}_k \in \mathbb{R}^{|B_k| \times d}$ be the activations corresponding to bucket $B_k$, noting that these can refer to nodes, edges or clusters, depending on the particular layer in the overall architecture. We can initially implement a linear layer by separately computing the matrix products for each category, as described in Alg. 2.

However, each time a matrix multiplication task is passed to the GPU, there is an overhead associated with transferring data and setting up the computation environment. If multiple smaller tasks are sent individually, this overhead is incurred repeatedly, which can reduce overall performance. Ideally, all tasks are combined into a single call, taking advantage of the GPU's architecture more effectively. This is why we propose a different approach for the mixed-precision matrix multiplications. Assume the bit-widths $b$ are given as multiples of a base-bit-width $b_0$, e.g., $b_0 = 4, b = (4, 8, 12)$ for `Int4`, `Int8`, `Int12` quantization. Given a quantization level $k$, bucket index $i$ and channel $c$ we then split the bitstring of $Q_k(\mathbf{a}_k^{ic})$ into $b_k/b_0$ bitstrings

---

[2]Although the auxiliary network's predictions may be random during the initial training phase, thus unsuitable for assessing regional complexity, we have observed that starting with a warm-up phase in which uniform weights $\mathbf{w}$ are used until $A$ is properly trained is unnecessary, not leading to better results.

---

**Algorithm 3 Optimized Mixed-Precision Linear Layer**

---

1: **Input:** Activations $\mathbf{a} \in \mathbb{R}^{N \times d}$, Weight matrix $\mathbf{W}^T \in \mathbb{R}^{d \times d'}$, Available quantization functions $Q = [Q_1, \ldots, Q_K]$, Bit-widths $b = [b_1, \ldots, b_K]$, Base bit-width $b_0$, Index buckets $B = [B_1, \ldots, B_K]$
2: $\{\mathbf{a}_k\}_{k=1}^K \leftarrow \text{Distribute}(\mathbf{a}, B)$ {Get $\mathbf{a}_k \in \mathbb{R}^{|B_k| \times d}$}
3: Initialize $\alpha \leftarrow [\emptyset]$ {Encoded activations}
4: Initialize Scales $\leftarrow [\emptyset]$
5: **for** $k = 1$ to $K$ **do**
6:     $Q_{\mathbf{a}_k} \leftarrow Q_k(\mathbf{a}_k)$ {Quantize activations in bucket $k$}
7:     $\alpha_k \leftarrow \text{Encode}(Q_{\mathbf{a}_k}, b_0)$ {Encode as $\texttt{Int-}b_0$ values}
8:     Scales $\leftarrow$ Scales $\cup \left\{ \frac{s_k}{s_w} \cdot 2^{m \cdot n} \mid m = 0, \ldots, \frac{b_k}{n} - 1 \right\}$ {Scale factors for each bit segment}
9:     $\alpha \leftarrow \alpha \cup \{\alpha_k\}$
10: **end for**
11: $\alpha \leftarrow \text{Concatenate}(\alpha)$
12: $S \leftarrow \text{diag}(\text{Scales})$
13: $Y \leftarrow S \cdot \alpha \cdot Q^w(\mathbf{W}^T)$ {Scaled matrix multiplication}
14: $Y \leftarrow \text{ScatterAdd}(Y)$ {Combine at original indices}
15: **Output:** $Y$

---

of size $n$. Omitting bucket and channel indices in $Q_k(\mathbf{a}_k)$ for ease of notation, we then redefine those $b_k/b_0$ smaller bitstrings as $\texttt{Int-}b_0$ values as follows:

$$Q_k(\mathbf{a}_k) = \sum_{i=0}^{b_k-1} 2^{\gamma_i}, \quad \gamma_i \in \{0, 1\} \tag{13}$$

$$\alpha_{km}(\mathbf{a}_k) := \sum_{i=mb_0}^{(m+1)b_0} 2^{\gamma_i - mb_0}, \tag{14}$$

where $m \in (0, \ldots, \frac{b_k}{b_0} - 1)$. We can thus encode $Q_k(\mathbf{a}_k) \in \{0, \ldots, 2^{b_k} - 1\}^{B_k \times d}$, which consists of $B_k$ vectors of $d$ channels with $\texttt{Int-}b_k$ precision, as $\alpha_k(\mathbf{a}_k) \in \{0, \ldots, 2^{b_0} - 1\}^{B_k \cdot \frac{b_k}{b_0} \times d}$. This flattened representation corresponds to $B_k \cdot \frac{b_k}{b_0}$ vectors of $d$ channels with $\texttt{Int-}b_0$ precision. Since this transformation applies to all $k$, we can combine all buckets into a single array $\alpha \in \{0, \ldots, 2^{b_0} - 1\}^{\tilde{N} \times d}$, where $\tilde{N} = \sum_k |B_k| b_k / b_0$. We can then perform a single matrix multiplication $\alpha Q^w(\mathbf{W}^T)$. Finally, we scale the output to reflect the effective contribution of each bit segment based on its position within the original encoded bit-string, while simultaneously applying existing scaling factors. The steps are summarized in Alg. 3.

## 5 Experiments and Results

Across all experiments, the auxiliary model is implemented as an MPNN with 3 hidden layers and 32 hidden channels. For Darcy and ShapeNet-Car, the main model is an MP-PDE with 6 hidden layers and 128 channels, containing about 550k parameters. For the EAGLE experiment, we employ GraphViT with a hidden dimension of 64 and 3 attention layers, totaling about 1M parameters. We implement our framework in $\texttt{flax}$ (Heek et al., 2024) and use the $\texttt{aqt}$ Research (2023) library for quantization. Further details are given in Appendix A and B. We run all experiments for all quantization regimes from $\texttt{Int4}$ until $\texttt{Int8}$, gradually increasing the fraction of $\texttt{Int8}$ nodes used for the mixed-precision runs. The main results are summarized in Figures 3 and 5. We also ran experiments at higher bit-width, but for brevity we only show results until the quantization level at which increasing precision further has no significant impact on accuracy. Note that the uniform curves show exponential decay for lower bit-width, with the convexity of the curves suggesting that it is nontrivial for combinations of categories to result in lower losses (i.e., plain averaging of points on a convex curve leads to higher values by definition, see also the random assignment in table 2).

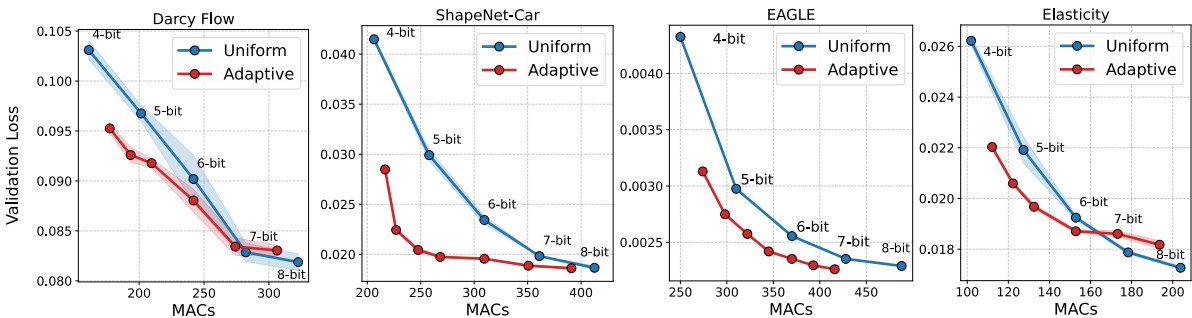

Figure 3: Validation loss vs cost (estimated in $10^9$ MACs) for uniform vs adaptive (ours) quantization. The latter is obtained through increasing the fraction of `Int8` nodes used, and includes the cost of the auxiliary model. Our framework demonstrates consistent Pareto improvements across all benchmarks. For Darcy, ShapeNet-Car and Elasticity, main models are MP-PDEs, for EAGLE - GraphViT.

While Figures 3 and 5 present results across a broad range of compute budgets, we note that practical interest is often concentrated on the more efficient (low-bit-width) low-cost models—corresponding to the left side of the curves, where MACs are lowest. The results indicate that our adaptive quantization method generally outperforms uniform quantization on the entire spectrum between `Int4` and `Int8`. However, in practice, uniform quantization between those endpoints is not implemented on most hardware at all. Therefore, a further benefit of our method is that it makes this full regime accessible in the first place. We refer to Appendix D for more details on hardware implementation.

## 5.1 ShapeNet-Car

**Dataset** The dataset is generated by Umetani & Bickel (2018) and consists of 889 car shapes from ShapeNet. Every car is represented by $N = 3600$ mesh points on its surface $\{\mathbf{p}_1, ..., \mathbf{p}_N\} \in \mathbb{R}^{N \times 3}$, and its aerodynamics is simulated by solving the Navier-Stokes equation. The task is thus to predict the pressure value $\mathbf{y} \in \mathbb{R}^N$ for every mesh node. We follow the preprocessing from Alkin et al. (2024) and use $k$ nearest neighbours to induce the connectivity. The train/test split contains 700/189 samples. Each (main) model is trained by optimizing the MSE loss between predicted and ground truth pressure.

**Results** As shown in Fig.3, our adaptive quantization achieves Pareto-optimal performance compared to uniform quantization. The approach works especially well in the low bit-width regime, matching the performance of 8-bit models while using only half the computational resources.

**Alternative auxiliary model** As an alternative to training the auxiliary model directly on the main model's loss, we experimented with a setup where the auxiliary model is independently pretrained on the data itself. We let the auxiliary model predict both a mean and a variance for each node, with the variance representing aleatoric (data-inherent) uncertainty in the sense of heteroskedastic regression (Kendall & Gal, 2017). We then use this variance to guide bit allocation, assigning higher precision to nodes with greater predicted uncertainty.

This approach relies on the assumption of a Gaussian likelihood model, which provides a tractable way to learn uncertainty but may not perfectly reflect the true noise distribution in the dataset. Nevertheless, it offers a reasonable proxy for node complexity without requiring joint training with the main model. When comparing this approach to our surrogate-loss method, we do not see a big difference when a large ratio of `Int8` is used. However, we expect this is because our assignment procedure (assigning `Int8` to the top-$k\%$ high-complexity nodes) hides differences in ordering for larger $k$. Indeed, we slightly favor the surrogate-loss method, as it yields better results for the more challenging lower resource budgets (Table 1).

Table 1: Validation loss for surrogate loss-based (default) and uncertainty-based bit-width assignment on the ShapeNet Car dataset. For the latter the auxiliary model is pre-trained on the data, and its variance is used as input to the bit assigner. While the difference is negligible when there are sufficient high-precision locations to distribute, it becomes clearer when only a small fraction is available that the default method assigns them better.

| Int4 / Int8 ratio | Uncertainty | Default |
|---|---|---|
| 95% / 5% | $0.0330 \pm 0.0042$ | $0.0285 \pm 0.0008$ |
| 90% / 10% | $0.0232 \pm 0.0004$ | $0.0224 \pm 0.0002$ |
| 80% / 20% | $0.0205 \pm 0.0002$ | $0.0204 \pm 0.0004$ |
| 70% / 30% | $0.0199 \pm 0.0002$ | $0.0197 \pm 0.0002$ |

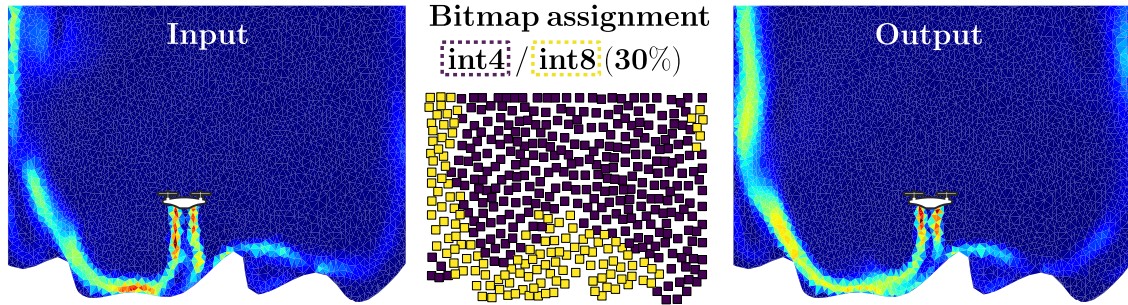

Figure 4: The proposed framework on an arbitrary example from the EAGLE dataset. The model predicts the field based on the PDE solution of the previous timestep. The bitmap assignment process manages to successfully capture the airflow and allocate resources to the relevant region. In this example 30 % of the nodes is assigned `Int8`, leaving the rest in low (`Int4`) precision.

## 5.2 Darcy flow

**Dataset** We use the dataset from Li et al. (2020) which contains the solution of the steady-state of the 2D Darcy FLow equation on the unit box. The data is defined on a regular grid $421 \times 421$, which we sparsify and treat as a directed graph with $N = 2770$ nodes. The task is to learn a map from the diffusion coefficient $\mathbf{x}_i \in \mathbb{R}_+$ to the solution $\mathbf{u}_i \in \mathbb{R}_+$ for node $i$. The (main) model is trained by optimizing the MSE loss between the predicted and ground truth solution.

**Results** The comparison between uniform and adaptive quantization is shown in Fig.3. The adaptive method improves quantization across all MACs tested, but less significantly so in the high MACs regime. We focus on the low bit-width regime, however, which is often of most interest in NN quantization research as this is where compute savings are highest and accuracy hardest to retain.

## 5.3 EAGLE

**Dataset** EAGLE (Janny et al., 2023) is a large-scale dataset ($1.1 \cdot 10^6$ meshes) resulting from non-steady fluid dynamics simulations. It consists of 600 unique scenarios of a moving flow source interacting with non-linear domains (see Fig. 4 for an example). Meshes are dynamic and vary in the number of nodes, with the average $N = 3388$. The dataset is especially challenging as the dynamics are often highly turbulent and multi-resolutional due to non-trivial environments. In particular, obstacle layouts vary between scenarios, so both training and validation include unseen configurations, providing an inherent out-of-distribution test for both models. The task is defined as follows: given the simulation state at time $t$ in the form of a graph $G^t$, predict the future pressure and velocity ($\mathbf{v}_i^{t+1} \in \mathbb{R}^4$) for every node after a time-step $dt$. The model is trained by minimizing the MSE on pressure and velocity (see Appendix A for details).

**Results** We observe a noticeable effect of adaptive quantization (see Fig.3). Again, adding only $\sim 10\%$ more MACs (by converting a small fraction of nodes to `Int8`) recovers more than half of the loss incurred

when quantizing from `Int8` to `Int4`. We also provide the example of the forward pass of our framework in Fig.4. Notice that the bit assigner manages to successfully capture the area of rapid changes along the borders. The main model can then allocate fewer resources to center regions with stable pressure and velocity while focusing on highly turbulent areas.

### 5.4 Elasticity

**Dataset**  The dataset is introduced by Li et al. (2023) and contains solutions to a 2D plane-stress hyper-elasticity problem with a central void. The domain is a unit square with a randomly-shaped hole, clamped at the bottom and subjected to vertical tension at the top. The material follows the incompressible Rivlin–Saunders hyper-elastic model. The data consists of unstructured point clouds with $N \approx 1000$ nodes per sample. Each sample consists of an input graph based on the spatial coordinates and an output graph based on the stress field at each node. The task is to learn a map from geometry to stress response, with models trained using MSE loss between predicted and true stress tensors.

**Results**  Here too, as shown in Fig.3, our adaptive quantization achieves Pareto-optimal performance compared to uniform quantization in the low bit-width regime. When only a small portion of `int4` nodes was used in the mixed-precision ratio, the loss increases relatively much: the addition of only a few `int4` hurts the training more than changing to `int7` precision. However, in practice, we expect interesting use cases to be on the opposite end of this setup: relatively few nodes getting assigned a high precision, with the bulk remaining in low precision.

### 5.5 Quantization assignment ablation

| Dataset | Targeted Assignment (default) | Random Assignment |
|---|---|---|
| ShapeNet Car | +3.4% | +14.6% |
| EAGLE | +2.6% | +37.2% |
| Darcy | +29.1% | +73.1% |
| Elasticity | +16.1% | +72.2% |

Table 2: Increase in loss as a result of quantizing half of the nodes to `Int4`, leaving the rest `Int8`. The default is to assign nodes with high (predicted) complexity to `Int8` quantization. For the random assignment, we shuffle the node complexities. The numbers are normalized such that +0 % corresponds to the "base" loss of the all-`Int8` uniform setup, and +100 % corresponds to the "fully quantized" loss of the all-`Int4` uniform setup. This way, random assignment of half the nodes to `Int4` resulting in $< 50\%$ loss increase indicates a relative robustness, whereas $> 50\%$ increase in loss indicates a disproportionately large sensitivity to quantizing nodes at `Int4`. Note that mitigating the increase in loss is all the more impressive for the latter category, where quantizing nodes to `Int4` so drastically hurts performance.

One might attribute the improved performance to dynamic quantization acting as a form of regularization. To test this effect, we conduct a control experiment (see Table 2) where we randomly shuffle the weights coming to the bit assigned during training, which keeps the stochastic behavior but removes focused resource allocation. Our results indicate that the focused allocation is key to the framework's performance.

While the improvements on Darcy and Elasticity may at first appear less striking than those for ShapeNet Car and EAGLE, it is in fact notable that the targeted assignment strategy substantially mitigates quantization loss in these cases. As shown in Table 2, random assignment leads to disproportionately large increases in loss, indicating that these datasets are highly sensitive to nodes being quantized to Int4. Against this challenging baseline, the much smaller increases under targeted assignment demonstrate that the method effectively preserves performance even where quantization is most damaging.

# 6 Discussion and Conclusion

We introduced Adaptive Mesh Quantization (AMQ), a general framework for improving the efficiency of quantized message-passing neural networks (MPNNs) by enabling spatially adaptive bit-width allocation. Bit allocation is guided using a lightweight auxiliary GNN that estimates the local prediction loss of the main model, thereby serving as a proxy for spatial complexity. This mechanism allows the model to concentrate computational resources where they are most needed, yielding a more efficient and targeted use of precision. AMQ consistently achieves favorable compute-performance trade-offs, outperforming uniform quantization baselines across diverse PDE surrogate tasks.

While larger datasets, i.e., meshes with more nodes, can be more practically relevant than the sizes often used in academic studies, we expect the benefits of AMQ to become even more pronounced in large-scale, high-resolution scenarios. Additional experiments on larger meshes (Appendix E) confirm that AMQ continues to perform beyond the size of small-scale research datasets. Nevertheless, we expect the benefits to be especially pronounced on real-world, high-resolution data, where spatial complexity is preserved in greater detail and the bit assigner can exploit local variation more effectively. Coarser meshes, in contrast, tend to average out complexity across regions, which can limit the effectiveness of adaptive schemes. This motivates exploration of AMQ in real-world, high-resolution simulation settings.

Our main results are reported in terms of MACs. To minimize potential discrepancies with real runtime, we implemented custom CUDA kernels and measured execution time directly (Appendix D). These experiments indicate that the relative trends are consistent, suggesting that MAC-based analysis provides a reasonable reflection of practical performance.

Finally, we note that further gains may be possible through task-specific hyperparameter tuning. Also, alternative strategies for complexity estimation—such as uncertainty-based metrics or learned priors—could complement or enhance the current loss-proxy approach.

# 7 Acknowledgements

This work is financially supported by Qualcomm Technologies Inc., the University of Amsterdam and the allowance Top consortia for Knowledge and Innovation (TKIs) from the Netherlands Ministry of Economic Affairs and Climate Policy.

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

# A    Training and architecture details

**MPNN**    Apart from the default size of 6 hidden layers and 128 channels, we also implemented the MPNN used in the Darcy and ShapeNet-Car experiments in a small variant (4 layers, 64 channels) and a large variant (256 channels). Results on these other variants are in 5. In each case, we use Adam (Kingma & Ba, 2015) with a weight decay of $1 \times 10^{-6}$, 500 epochs, and normed gradient clipping of 1.0. For Darcy and Elasticity, learning rates of $1 \times 10^{-3}$ and $5 \times 10^{-4}$ is used, respecitvely, and a batch size of 16. For ShapeNet-Car, the learning rate is $1 \times 10^{-3}$ and the batch size is 8. The (train and validation) loss in Darcy is the relative norm, as in Li et al. (2020), and for Elasticity and ShapeNet-Car, we use MSE. For all meshes, we use K-nearest neighbors to get edges, with $K = 5$, self-loops included, and $K = 10$ for Elasticity. Mean aggregation is used to aggregate message features. Furthermore, a cosine learning rate decay with linear warmup of 5 epochs is used.

**GraphViT**    For the GraphViT, we implemented quantized variants of each of the original modules used in Janny et al. (2023). We use 64 hidden channels for all node and edge-related operations and 128 hidden channels for the cluster state space. We apply 3 message passing layers before the cluster processing layers. These update both the nodes and edge features, with the original edge features being defined as the norm and relative distance between nodes. After the cluster processing, which is done using a gated recurrent unit to obtain cluster features followed by self-attention (see Janny et al. (2023) for details), a final message-passing layer is applied on the nodes that have the cluster features appended to them. We use a cosine learning rate decay schedule with 10 linear warmup epochs and a $K$-means clustering algorithm with $K = 400$. We train for 1000 epochs, using Adam (Kingma & Ba, 2015) with weight decay of $1 \times 10^{-6}$, normed gradient clipping of 1.0, learning rate $5 \times 10^{-4}$ and a batch size of 8. The edges are provided with the EAGLE dataset. We also train a large variant that uses 96 channels for nodes and edges and 192 for the clusters, as well as a small variant using only 2 message-passing layers before the cluster processing, 32 channels for the message-passing layers and 64 channels for the clusters. Results on these other variants are in 5. For each network, the (train and valid) loss is MSE, with a factor of $\alpha = 0.1$ applied to the pressure channels, as in (Janny et al., 2023).

**Auxiliary model**    The auxiliary model is a simple message-passing GNN with 3 hidden layers of 32 channels. Its inputs are the same as the main model, and it has one output channel representing the proxy measure of prediction complexity in each node. Training settings for the auxiliary model are identical to those for the main model that it operates with. The target is the normalized, detached loss of the main model, smoothened using 10 graph diffusion steps. Each graph diffusion step is implemented as

$$L_i^{t+1} = \frac{L_i^t}{2} + \frac{1}{2} \sum_{j \in \mathcal{N}(i)} L_j^t. \tag{15}$$

# B    Quantization

We implement quantization using the default settings of the `aqt` Research (2023) library. This means that the scale factor is determined per channel, based on maximum absolute value calibration, there are no learned quantization parameters, and activations are quantized symmetrically. We do note that this is not optimal because our frequent use of the GELU activation function means most activations are positive or relatively small when negative. However, the effect of finding a better quantization strategy would apply to both the uniform and the adaptive experiments. To keep things simple, we decided to stick with the default settings, leaving general improvements open for future work.

We did not optimize quantization for speed or accuracy because the latest and developing techniques in that field are mostly orthogonal to our approach. Also, the quantization library `aqt` we used only *simulates* the reported bit-width while executing in `Int8` on hardware. Consequently, we report MACs rather than latency measurements, as the latter would not accurately reflect the potential gains from optimized hardware implementation. For the uniformly quantized runs, we do note that intermediate bit-widths like `Int5` do not exist on hardware in practice, making our combined approach of `Int4`/`Int8` already more hardware-friendly. Our primary goal is to demonstrate theoretical feasibility, with hardware-specific optimizations

being beyond our current scope—particularly as many such optimizations are complementary to our core contribution. Nonetheless, for completeness, an indicatory analysis on runtimes is given in Appendix Section D.

Table 3: Validation loss for different auxiliary model sizes across various `Int4`/`Int8` ratios for MPNN on ShapeNet-Car.

| `Int4`/`Int8` ratio | Default | Small | Tiny |
|---|---|---|---|
| 90% | 0.0239 | 0.0232 | 0.0230 |
| 80% | 0.0207 | 0.0206 | 0.0207 |
| 70% | 0.0201 | 0.0197 | 0.0196 |
| 50% | 0.0192 | 0.0193 | 0.0190 |

Table 4: Validation loss for different auxiliary model sizes across various `Int4`/`Int8` computation ratios for GraphVit on EAGLE.

| `Int4`/`Int8` ratio | Default | Tiny |
|---|---|---|
| 80% | 0.0028 | 0.0031 |
| 50% | 0.0024 | 0.0023 |

## C   Results with different model sizes

To test the generalizability of our framework across different main model sizes, we reproduce the experiments from the main body and reduce/increase the number of MACs considerably. The results are shown in Fig. 5. Architecture details of small and big variants are in the previous Appendix Section A. Overall, we reproduce previous conclusions and demonstrate that the performance benefits are preserved across scales.

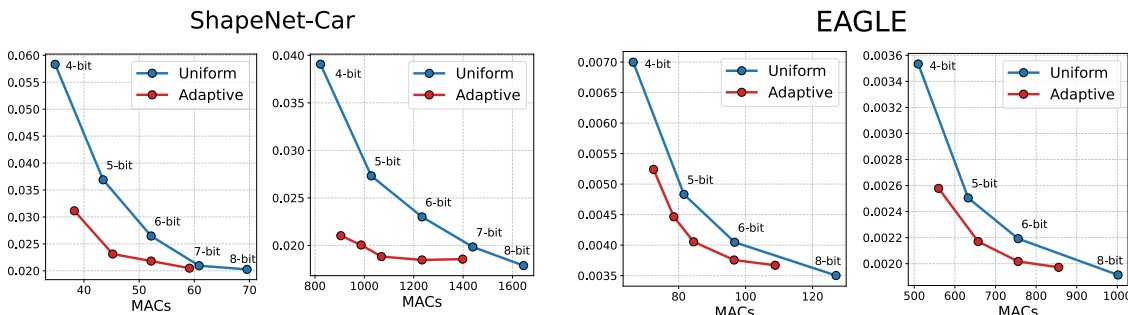

Figure 5: Performance of smaller (**left**) vs larger (**right**) main model on ShapeNet-Car and EAGLE tasks.

Additionally, we test how the size of auxiliary models affects the quantization result (see Table 3). We introduce a small and tiny variant: The small variant uses `Int4` quantization in the auxiliary model, instead of `Int8`. The tiny variant also uses 24 features and 2 hidden layers instead of 36, 3 respectively. We found the framework to be robust to the design choice. Overall, even a tiny model is sufficient for bit-width allocation, which is beneficial as its inference provides little to no overhead. For completeness, Fig. 6 shows the overhead of the Auxiliary model compared to other layers in the GraphViT and the MPNN model that we used, indicating its negligible overhead.

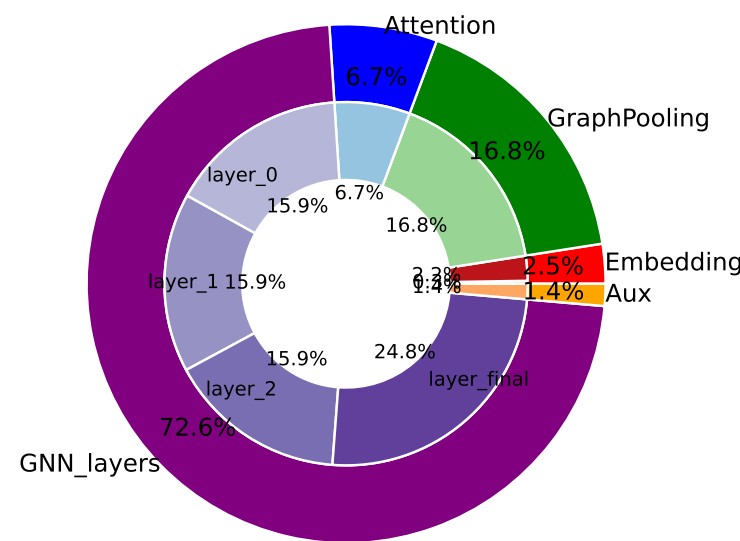

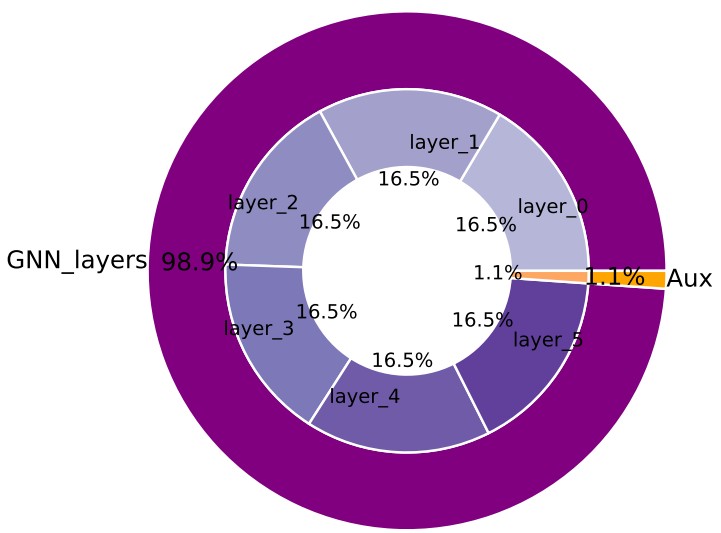

Figure 6: Distribution of MACs for a default size GraphViT model on EAGLE dataset (top) and MPNN on Darcy dataset (bottom). The method demonstrates robustness to the size of the auxiliary model, with even minimal overhead sufficient for guiding effective bit allocation.

# D    Runtime analysis

Because training costs of Neural Operators can be amortized (train once, deploy on many different PDE initial conditions) we conduct runtime analysis by focusing on inference, assuming weights are already quantized. Our analysis targets matrix multiplications, which dominate the cost, and because other identical pipeline components cancel when comparing mixed precision with uniform quantization. While compute generally scales quadratically with matrix size and memory traffic scales linearly, the latter can still become the bottleneck as compute capability of modern GPUs keeps increasing. To investigate to what extend MACs is a reasonable measure of efficiency, we profiled custom CUDA kernels (based on CUTLASS) for our mixed-precision GEMMs using NVIDIA Nsight Compute.

Quantized GEMM can be decomposed into: (i) quantization of activations (weights pre-quantized), (ii) GEMM, and (iii) dequantization of outputs. Kernel runtime depends on read, write, and compute operations. Since dequantization outputs are typically `Int32`, writes are costly and in practice GEMM and dequantization are often fused. However, because such customization options are limited for low-bitwidth GEMMs in CUTLASS we implemented quantization, GEMM, and dequantization separately for both uniform and mixed-precision variants. We approximate fused runtime by omitting redundant writes and reads via null pointers and dummy indices.

For runtime evaluation, we evaluate a range of settings designed to represent realistic deployment scenarios. In particular we assume inputs and outputs are float16, weights are already given and quantized to `Int4`. Low-precision nodes use int4 activations, high-precision nodes `Int8`. While mixed settings rely only on `Int4`×`Int4` kernels and custom quantization/dequantization, uniform runtimes cannot be fully measured since no `Int5-7` kernels exist in hardware. This makes our ability to operate between `Int4` and `Int8` an inherent advantage. For comparison we linearly interpolate uniform runtimes between `Int4` and `Int8`. We use 512 input/output channels and vary node counts from 4096 to 65536.

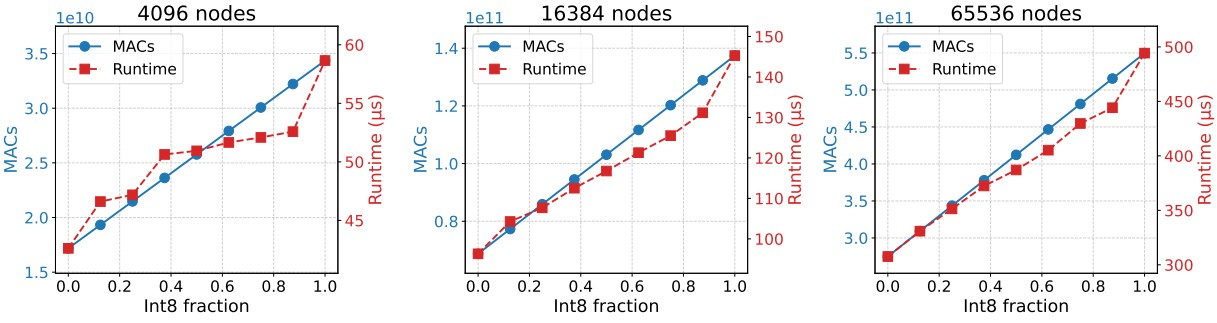

Figure 7: Runtime results vs. MACs. At 0% or 100% int8 the setup is uniform; intermediate points are mixed-precision. Red squares represent the run-time results, and the corresponding blue circles their respective MAC. The different $y$-axes are scaled so that coincident (red and blue) lines would imply a linear relationship. There are no uniform implementations between the two extremes, but if for comparison purposes one assumes a linear trend from the `Int4` to the `Int8` measurement, this would correspond to the blue (MACs) line. We find that MACs appear to be a reasonable, in some cases even slightly conservative, estimate of runtime savings.

# E    Results on larger meshes

As an ablation to test our method on larger-scale problems, we ran experiments using MPNN on two extra datasets:

- ShapeNet Car Large Umetani & Bickel (2018), containing 31,186 nodes. It is different from our default variant, as it also includes the air around the car. The model is trained to predict both the velocity and pressure in this significantly larger larger 3D domain.

- Darcy Large. This is a higher resolution variant of the Darcy dataset used in the other experiments, containing 11236 nodes. As we did not proportionally scale the model, in particular using the same number of neighbors per node, the task is significantly harder.

Table 5 presents results for three key quantization configurations that represent the most practically relevant scenarios. These configurations correspond to hardware-implementable quantization levels, as intermediate bit-widths (`Int5-Int7`) are typically unavailable.

| | ShapeNet Car Large | | Darcy Large | |
|---|---|---|---|---|
| Configuration | MACs | Loss | MACs | Loss |
| `Int4` (uniform) | 1619 | $0.0226 \pm 0.0002$ | 645 | $0.132 \pm 0.0002$ |
| 90% `Int4`, 10% `Int8` | 1781 | $0.0167 \pm 0.0007$ | 710 | $0.125 \pm 0.0003$ |
| `Int8` (uniform) | 3239 | $0.0103 \pm 0.0002$ | 1290 | $0.120 \pm 0.0017$ |

Table 5: Performance comparison on large-scale datasets. The adaptive mixed-precision setup recovers roughly half of the accuracy loss from `Int8` to `Int4` by converting just 10% of nodes to `Int8`.

The results demonstrate the effectiveness of our adaptive approach on larger problem sizes. The mixed-precision configuration shows that strategic allocation of higher precision to a small fraction of nodes can bridge much of the performance gap between uniform `Int4` and `Int8` quantization, while maintaining computational efficiency close to the `Int4` baseline.

