# OpenReview forum: "Adaptive Mesh Quantization for Neural PDE Solvers"
_TMLR — Accepted by TMLR_

### Review · Reviewer_n9Fj · 2025-08-22

**Summary Of Contributions:**

The high-level idea of this paper is straightforward: it aims to develop an efficient neural PDE solver by adaptively adjusting the quantization to accommodate the possibly different computational needs across various regions in a fixed mesh topology. The perspective of this paper largely focuses on advocating the power of quantization models, which is different from (and complements) many existing neural PDEs that advertise new network designs, algorithms, or data. The paper evaluates its method on typical PDE test problems in scientific machine learning and reports its performance against uniformly quantized baselines.

**Audience:**

Yes

**Audience Explanation:**

The paper tells an interesting and, as far as I know, novel story. Incorporating quantization methods into neural PDEs is an underexplored topic, and the paper makes a timely contribution in this regard. I believe this combination is not the end of an idea but the start of several interesting follow-ups.

**Broader Impact Concerns:**

None.

**Claims And Evidence:**

Yes

**Claims Explanation:**

The technical method looks sound to me at a high level. I am not an expert in quantization models, so I will leave it to other expert reviewers.

The experiments are convincing enough to showcase the proposed adaptive quantization model’s benefits over a uniformly quantized one. The experiments also cover a reasonably diverse set of typical PDEs in fluid and solid mechanics.

**Requested Changes:**

I will start with two high-level comments on the paper’s baselines and scalability, respectively.

Baselines: The storytelling in the introduction emphasizes the shortcomings of adaptive mesh refinements (AMR) and uses them to motivate the proposed AMQ. Surprisingly, the paper does not plan to compare them (“While our method is inspired by and conceptually related to AMR, we do not aim for a direct comparison”). I am not entirely sure whether excusing the paper from such a comparison is fair to AMR, and I would like to be convinced by a simple motivating example that can present the pros and cons between AMR and AMQ, highlighting the paper’s claimed motivation from AMR to AMQ. Alternatively, I’d also be okay with an overhaul of rewriting the abstract/introduction to completely remove AMR from the story.

Scalability: Despite the positive results across various PDE problems, their problem sizes remain relatively small: The test problems contain at most ~3k node numbers, most of which are in 2D only. For a 3D problem, 3k nodes would be insufficient even for a coarse grid resolution of 20 x 20 x 20. Therefore, it is unclear to me whether the proposed model can be scaled up to a practically useful level. I appreciate that the paper has commented on this topic in the “Discussion and Conclusion” section, but I still would like to bring this up and welcome any additional experiments.

The following changes will have critical effects on my evaluation of this paper. The goal of these changes is to help us better understand the proposed method’s properties:
1. Comparing AMR with AMQ in a simple, motivating example and discuss their pros and cons. See “Baselines” above.
2. Select one experiment, scale its problem size up until the proposed method struggles to learn an effective quantization model, and discuss the results. See “Scalability” above.

I would also like to request a few text changes:
1. The title “adaptive mesh quantization” looks confusing to me because meshes are actually fixed in the proposed method. Please consider renaming or clarifying it.

2. “Compared to models working on regular grids, they [graph neural networks] do not require interpolation, thereby reducing both cost and error.” I cannot see how the difference between regular grids and meshes (graphs) can lead to the conclusion that GNNs do not require interpolation. Please justify.

3. “In certain domains, such as molecular data or graph-based representations of real-world systems, nodes represent specific, meaningful points (e.g., atoms in a molecule or locations in a sensor network) that cannot trivially be moved.” The molecular data argument here is a bit contrived because it actually does not apply to the paper’s problem setup (PDEs in fluid/solid mechanics). I suggest replacing it with a better motivation for quantization models.

---

> ### Author Response · Authors · 2025-09-26
> **Storyline on Adaptive Mesh Refinement**
>
> We thank the reviewer for this feedback and have chosen to rewrite the abstract and introduction to remove the emphasis on AMR:
> - Our original framing incorrectly suggested that AMR is a well-established method in neural PDE surrogates; upon further research we find this is not the case. While non-uniform meshes are commonly used, they are typically prepared in advance based on geometric considerations or data-specific heuristics, rather than being adaptively refined based on predicted modeling difficulty during inference. Implementing a proper comparison may then require developing this method specifically for our neural PDE surrogates and data, which we consider an extension of our current work, rather than a necessary comparison.
> - Our revised storyline focuses more on the mismatch between where computational precision is needed (based on physical complexity) and where computational resources are normally allocated (uniformly across all mesh nodes). Irregular meshes naturally arise from geometric constraints and boundary conditions, but the regions requiring higher numerical precision for accurate PDE modeling do not have to align with this geometric structure. For instance, turbulent phenomena or steep gradients can occur in geometrically simple regions, while complex boundaries may have relatively predictable behavior.

---

> > ### Comment · Reviewer_n9Fj · 2025-09-29
> >
> > OK. Sounds good to me.

---

> ### Author Response · Authors · 2025-09-26
> **Scalability**
>
> We tested AMQ on two larger datasets: a ShapeNet Car variant with 32.2K nodes and a Darcy variant with 11.2K nodes, with results in Appendix E. These correspond to scaling factors of 4× and 9× relative to the original datasets, but our method did not break and further increases beyond these sizes were limited by GPU memory. We do find that AMQ continues to perform well in this larger regime: in both cases adding only ~10\% more MACs (by converting a small fraction of nodes to int8) recovers roughly half of the accuracy loss incurred when quantizing from int8 to int4.

---

> > ### Comment · Reviewer_n9Fj · 2025-09-29
> >
> > Thank you for the clarification.
> >
> > "...but our method did not break and further increases beyond these sizes were limited by GPU memory."  This is good to know. It would be great to mention it in the Discussion and Conclusion section.

---

> ### Author Response · Authors · 2025-09-26
> **Text changes**
>
> ### Title
> *"The title “adaptive mesh quantization” looks confusing to me because meshes are actually fixed in the proposed method. Please consider renaming or clarifying it."*
> While rewriting the abstract, we took the opportunity to clarify the title in a more explicit way: although the mesh itself remains fixed, the term “adaptive” refers specifically to the quantization applied to mesh elements—nodes, edges, and potentially other meta-features such as clusters. In the abstract, we explicitly introduce “Adaptive Mesh Quantization” and immediately define it as spatially adaptive quantization across mesh nodes and edges, making clear that the adaptivity applies to the quantization, not the mesh structure itself. We would also be happy to hear the reviewer’s opinion on whether adding a hyphen (“Adaptive Mesh-Quantization”) might improve clarity. We also included an extra paragraph when rewriting the introduction, further clarifying the title, i.e., that "adaptive" refers to quantization of the mesh, not to the mesh itself:
> *"We address this inefficiency by introducing Adaptive Mesh Quantization (AMQ). In mesh quantization, we
> apply quantization to all the mesh-based features - including node features, edge features, and possibly
> cluster representations - processed by graph neural networks operating on meshes. Our adaptive approach
> dynamically varies this quantization - allocating higher bit-width to regions of higher complexity."*
>
> ### GNN interpolation
> *"I cannot see how the difference between regular grids and meshes (graphs) can lead to the conclusion that GNNs do not require interpolation. Please justify."*
> We appreciate the reviewer’s comment highlighting this imprecision; the original phrasing was indeed misleading, as interpolation may still be required between meshes of different resolutions. Our intended point was that, unlike grid-based neural models like CNNs, GNNs do not require to map mesh data onto a regular lattice. We have revised the sentence as follows:
> "Graph Neural Networks (GNNs) have emerged as a natural solution, since they can operate directly on unstructured meshes, thereby avoiding the mesh-to-grid interpolation otherwise needed to represent irregular domains on regular grids."
>
> ### Better motivation for AMQ of nodes
> *"3. The molecular data argument does not apply to the paper’s problem setup (PDEs in fluid/solid mechanics). I suggest replacing it with a better motivation for quantization models."*
> We thank the reviewer for this comment. Although indeed not reflected in our experiments, we had included this argument because GNNs, and our method in particular, still apply in such domains, where node positions are fixed by definition and adaptive quantization allows flexibly allocating MACs. That said, for more general PDE settings, we can make the point that the mesh primarily captures geometry (boundaries, interfaces), while the regions that are most computationally challenging may not necessarily align with this geometric structure (e.g., turbulence, shocks). Adaptive quantization addresses this by allocating precision where PDE modeling is most difficult, even on a fixed irregular mesh. We included this reasoning as part of our rewrite of the introduction.

---

> > ### Comment · Reviewer_n9Fj · 2025-09-29
> >
> > Thank you for the text editing. I have no further questions in this regard.
> > I appreciate the response in GNN interpolation.

---

### Review · Reviewer_uJSi · 2025-08-22

**Summary Of Contributions:**

This paper proposes an approach that adaptively allocates bit precisions to different grid resolution and improves computation efficiency.

**Audience:**

Yes

**Audience Explanation:**

Reducing computation cost is an important problem in Neural PDE methods.

**Broader Impact Concerns:**

None.

**Claims And Evidence:**

Yes

**Claims Explanation:**

The proposed algorithm is tested on several large-scale problems, and the methodology seems sound to me.

**Requested Changes:**

+ In this work, the computation cost is measured by the total number of MACs executed during the training. While I agree with the authors view that wall-time is hardware-dependent, it feels that this metric is a bit obsolete. Actually, there are other aspects which may consume a significant amount of time, such as data reading, new model training and evaluating and GPU/CPU switching, that are rarely concerned in traditional methods. Therefore, it would be much more convincing to show that the proposed method achieves better wall-time efficiency, not just MAC efficiency.

+ A minor concern: as shown in Figure 3, the validation loss for uniform and adaptive methods is not significant as they are on the same magnitude. I am fine with the results because the core contribution of this work is not the performance improvement, but a bit curious why the curve of Adaptive stops exactly at the intersection with the Uniform curve in the third and fourth plot. It seems that the Uniform curve will go under the Adaptive curve if MACs keep increasing.

+ I recommend the authors to release the code upon the acceptance of this work.

---

> ### Author Response · Authors · 2025-09-26
> **Compute cost measurement**
>
> ### Wall time vs MACs
> Precise runtime measurements for our full pipeline is challenging as efficient low-bitwidth support is limited: Many quantization frameworks typically simulate quantization and revert to int8 under the hood, reflecting that quantization research often only focuses on accuracy at simulated bitwidths, leaving efficient implementations aside. Our work does consider hardware feasibility by proposing an algorithm that allows efficient GPU implementation, though achieving full runtime benefits would require substantial custom kernel engineering. (While some basic int4×int4 kernels exist (e.g., in CUTLASS), they allow little freedom to customize further.)
>
> Nevertheless, we took the reviewer’s concern seriously and designed a realistic proxy to approximate runtime. Using CUTLASS, we implemented separate quantization, GEMM, and dequantization kernels, and removed redundant memory operations (writes from GEMM, reads for dequantization) to mimic a fused implementation. This gives a useful estimate of runtime efficiency.
>
> Our measurements confirm that MAC counts are a reasonable estimate of runtime when comparing mixed-precision and uniform quantization. Thus, our results are expected to lead to real runtime improvements when implemented. These results are included in the appendix, and we will release the C++ code.
>
> ### Other aspects
> *"there are other aspects which may consume a significant amount of time"*
> For aspects not covered in our analysis, we argue that these costs are either amortized (e.g., a network is trained once and then applied to many different initial conditions, hence the common focus on inference time for neural PDE surrogates) or cancel out when comparing uniform versus mixed-precision experiments (when both incur the same costs). We have clarified this reasoning in the extra runtime section.

---

> ### Author Response · Authors · 2025-09-26
> **Result curves**
>
> ### Uniform curve vs Adaptive curve at high MACs
> *"It seems that the Uniform curve will go under the Adaptive curve if MACs keep increasing."*
> Most points in Figure 3 were obtained by step-wise increasing the fraction of int8 nodes from 0 (all int4) to 1 (all int8). Consequently, if such curves continue to the uniform int8 point, mixed-precision runs would no longer be mixed, and the curves necessarily converge.
> That said, we indeed observe (and confirm with new experiments) that mixed-precision is slightly outperformed by uniform setups near the end of the curve. This may occur because adding even a small fraction of int4 nodes can have more impact than reducing bit-width uniformly from int8 to int7. However, this regime is not a typical use case, so the difference is not practically significant: making a very small fraction of nodes int4, leaving the rest int8, is likely less practical than the other way around. We mainly included larger bit-widths only for presentation, as it shows the exponential increase in loss when quantizing to lower bitwidth. We updated the results paragraph for Elasticity to include the above notes.
>
> ### General presentation
> Admittedly, Figure 3 did not clearly convey that mixed-precision curves result from varying (e.g., int4/int8) ratios. The presentation was somewhat confusing because we had a few points using a different setup (int8/int16) but only indicated MACs. Since all reviewers agree that the claims made in the submission are supported by our evidence, i.e., that mixed-precision outperforms uniform runs, we focused on improving clarity rather than including every possible result.
> We made the following changes:
> - Added more runs and included error bars (for points where uncertainty region is sufficiently large to be visible)
> - Unified the domain and setup across datasets (starting at int4, ending at int8). In practice, low-bitwidth regimes are typically the main focus in NN quantization.
> - Optimized Darcy hyperparameters for more consistent results (tuned to get optimal uniform-precision runs).
> - Added a 90% int8 / 10% int4 case for each dataset to clarify behavior at the high-MAC end of the curve.
>
> We believe the revised figure is clearer, and we appreciate the reviewer’s comment that motivated this improvement. Naturally, the code will be released upon acceptance of this work.

---

### Review · Reviewer_Wnvc · 2025-09-17

**Summary Of Contributions:**

The authors introduce AMQ, a jointly trained loss-guided bit-width scheduler for GNN-based PDE solvers that adaptively ups precision in hard regions under a fixed compute budget, with an efficient mixed-precision implementation. The method is evaluated on several high-dimensional data sets, with generally encouraging results and improvements over uniform quantization at matched MACs.

**Audience:**

Yes

**Audience Explanation:**

The paper is very well written and strikes a good balance between being self-contained and assuming background expertise. The topic is timely; adaptive precision for neural PDE solvers/graph operators is of broad interest to the ML-for-Science, systems/quantization, and GNN communities.

**Claims And Evidence:**

Yes

**Claims Explanation:**

The empirical results show Pareto gains over uniform quantization at matched MACs across four tasks and two architectures, and include ablations (random/low-weight assignment; surrogate-loss vs uncertainty proxy). That said, the presentation can be improved and some additional experiments can provide a clearer picture of the results (see requested changes). Additionally, limitations of the method need to be discussed.

**Requested Changes:**

- The notation can profit from a general proofreading to ensure consistency. For instance, bold-font should be use consistently to denote vectors, including complexity weight vectors (**Algorithm 1**). Allocation ratios should be constrained to be positive. Matrices should be denoted either with an upper or a lower case, but not interchangeably so (consider also using bold font for matrices in keeping with standard notation). Some typos are also present, especially in the **Experiments and Results** section.
- I feel that the role of the smoothing function in Eq. 7 needs to be explained in more detail.
- $\mathcal{Q}$ in Eq.9 was never defined previously, but was simply referred to as $Q$ (see point on consistency).
- $B$ from **Algorithm 1** was redefined to $\beta$ in Eq. 9.
- Instead of fixing the network hyperparameters in all experiments, it would have been helpful to define model categories (e.g., "tiny", "base", "large"), as is typical in many DL fields and further explore performance as a function of model size.
- Why do some of the loss curves in Figure 3 display a small uncertainty region whereas other do not?
- Table 1 should definitely subdivide the results per data set and use NRMSE for comparison in case the outcomes have different scales. It should also show the loss for the adaptive method (and quantization ratio used) and specify that this is indeed validation loss. What is the error computed over currently? Data sets? Different model runs? Bootstrap?
- The results displayed in Figure 4 (EAGLE) data set are impossible to interpret for non-experts. How was the input selected? What does the 30% refer to?
- The surrogate uncertainty loss comparison needs a bit more motivation and explanation. It is important to note that this is aleatoric uncertainty coming from a heteroskedastic model, as described in Kendall & Gall (2017), and its quality is strongly dependent on the Gaussian assumption being legitimate.
- There are no limitations of the method mentioned. Consider discussing these. Here is what I think and I am happy to discuss with the authors:
1. The need to train an auxiliary model which may be difficult to train for complex inputs / low data settings).
2. If spatial complexity shifts into an out-of-distribution (OOD) regime (e.g., new obstacle layouts), does the bit assigner still focus on the right regions? Consider adding a held-out scenario split for EAGLE and demonstrating a case where the method breaks.
3. Did I understand correctly that all weights are fixed at Int8 and only activations are quantized? If that's the case, I would discuss differences between adaptive activations vs adaptive weights+activations and possibly provide an ablation to justify the choice.
4. AQT simulates bit-widths, but in real hardware, MACs does not equal latency, as the authors point out. Perhaps a reiteration of this point in the discussion would be helpful.

---

> ### Author Response · Authors · 2025-09-26
> **Response to reviewer Wnvc: main points**
>
> ### Notation fixes
> We thank the reviewer for spotting these and have incorporated the requested changes & proofread the paper.
> ### Uncertainty region in loss curves
> *"some of the curves in Figure 3 display a small uncertainty region whereas others do not"*
> We now include uncertainty in all curves (for points where uncertainty region is sufficiently large to be visible). We also used this opportunity to improve the overall presentation (see our related response to reviewer uJSi).
> ### Model sizes
> *"define model categories ("tiny", "base", "large") and further explore performance as a function of model size"*
> We include experiments on tiny (2/3 layers, 1/2 channels) and large (twice the channels) variants of MPNN and Graph-ViT in Appendix C, (Fig. 5) reproducing the same conclusions.
> ### Smoothing function
> *"role of the smoothing function in Eq. 7 needs to be explained in more detail"*
> We clarified the description (after Eq 7) by removing an unnecessary step and using the space to more clearly explain how normalization and graph diffusion stabilize training and reduce outlier effects.
> ### Table 1 Subdivide results per dataset
> *"Table 1 should subdivide the results per data set"*
> The original table used only ShapeNetCar with 25% int8, uncertainty measured over multiple model runs. We changed it and extended this ablation to the other datasets. To enable comparisons across datasets, we now report the resulting loss as relative increase, normalized by the difference between the "fully quantized" loss of the all-int4 uniform setup and the "base loss" of the all-int 8 setup. The table (repeated below) shows how Shapenet Car and EAGLE are relatively robust, whereas Darcy and Elasticity are relatively sensitive to quantizing nodes to int4.
> | Dataset   | Targeted Assignment (default) | Random Assignment |
> |--|--|--|
> | ShapeNet Car | +3.4% | +14.6%|
> | EAGLE | +2.6% | +37.2%|
> | Darcy | +29.1% | +73.1%|
> | Elasticity | +16.1% | +72.2%|
>
> *Increase in loss when quantizing half the nodes to int4 (rest int8).
> Values are normalized so that +0% = all-int8 baseline, +100% = all-int4 fully quantized, implying that random assignment leading to  <50% loss increase indicates relative robustness, whereas >50% indicates relatively large sensitivity to this quantization.*
>
> Results confirm our methods' effectiveness compared to possible regularization from random assignment.
> ### Figure 4 extra explanation
> *"Figure 4 (EAGLE) data set impossible to interpret for non-experts."*
> The input in Figure 4 is an arbitrary example from the EAGLE dataset. “30\%” indicates nodes at Int8; the remaining 70\% are Int4. We have updated the caption for clarity.
> ### Uncertainty loss extra explanation
> *"surrogate uncertainty loss comparison needs a bit more motivation and explanation"*
> We thank the reviewer for this input and improved the section. Indeed, the Gaussian assumption is not perfect, but we chose it as a convenient, simple proxy. This alternative approach is notable mainly because it can be trained purely on data and applied independently of the main model, which is the contrast we wanted to highlight.

---

> > ### Author Response · Authors · 2025-09-26
> > **Response to reviewer Wnvc: further discussion points**
> >
> > ### Need to train auxiliary model
> > *"the need to train an auxiliary model which may be difficult to train for complex inputs/low data settings"*
> > Our experiments show that the surrogate model can be much smaller than the main model. If task complexity increases, this likely also requires a larger main model, in which case the surrogate model can scale proportionally. In low data settings, the main model is more likely to be the performance bottleneck, and it may still exhibit predictable errors for the surrogate to detect, justifying the allocation of a fraction of MACs to it.
> > ### Out of distribution regime
> > In the EAGLE dataset, obstacle layouts already vary between training and validation, so both the main and auxiliary models encounter unseen configurations at test time. We further highlight this in the dataset description for clarity.
> > ### Weights not adaptive
> > *"Weights are fixed and only activations are adaptively quantized"*
> > Adaptive weight quantization would either require quantizing them during inference, or multiple prequantized weight sets. The first option would add a latency contribution that is usually explicitly avoided in quantized NN inference, whereas the latter is akin to a Mixture-of-Experts approach: one can then also use completely different weights altogether. This is a potentially valuable direction, but it is beyond the scope of this work.
> > We focus on activations because commonly the weights can be much more easily quantized to low bit-width, (int4 being Pareto-optimal) while it is precisely reducing activations to similar precision that hurts performance.
> > ### MACs vs Latency in real hardware
> > *"in real hardware, MACs do not equal latency"*
> > We added a section in the appendix on runtime analysis, where we find proportionality in an exemplary toy setup.  We also note that in real hardware intermediate bitwidths (int5–7) are unavailable, so the ability of our method to operate in the regime between int4 and int8 is an additional strength.

---

### Decision · Action_Editor_Fw2m · 2025-10-25

**Recommendation:** Accept as is

**Audience:**

Yes

**Audience Explanation:**

For the sake of transparent decision-making, I include here TMLR's criteria for acceptance that each accepted paper must meet:

- Are the claims made in the submission supported by accurate, convincing, and clear evidence?
- Would at least some individuals in TMLR's audience be interested in knowing the findings of this paper?

Reviewers agree that the paper meets the second criterion mentioned above related to "Audience", and some even believe that it goes further by making a meaningful contribution with the potential for real-world impact.

**Claims And Evidence:**

Yes

**Claims Explanation:**

For the sake of transparent decision-making, I include here TMLR's criteria for acceptance that each accepted paper must meet:

- Are the claims made in the submission supported by accurate, convincing, and clear evidence?
- Would at least some individuals in TMLR's audience be interested in knowing the findings of this paper?

All reviewers agree that the paper clearly meets the first criterion related to "Claims And Evidence" by providing claims that are sound and well supported by evidence. The reviewers raised some questions and concerns, which the authors have adequately addressed in the revisions to the reviewers' satisfaction.